# Transcriptome Analyses Provide Insights into the Auditory Function in *Trachemys scripta elegans*

**DOI:** 10.3390/ani12182410

**Published:** 2022-09-14

**Authors:** Ningning Lu, Bo Chen, Jiao Qing, Jinhong Lei, Tongliang Wang, Haitao Shi, Jichao Wang

**Affiliations:** Ministry of Education Key Laboratory for Ecology of Tropical Islands, Key Laboratory of Tropical Animal and Plant Ecology of Hainan Province, College of Life Sciences, Hainan Normal University, Haikou 571158, China

**Keywords:** *Trachemys scripta elegans*, transcriptome, inner ear, tympanic membrane, hearing sensitivity

## Abstract

**Simple Summary:**

Auditory function is an important sensory ability that contributes to the survival and reproduction of vertebrates. Studies have shown that turtles can hear and that sex-related differences exist in the auditory function of *Trachemys scripta elegans*. However, the associated gene expression characteristics are unknown. Therefore, we performed comparative transcriptomics to identify hub genes related to hearing organs involved in development and signal transduction. Six differentially expressed genes in the GABAergic synapse pathway were identified to explain the differences in hearing sensitivity. These results offer new insights into the genetic mechanisms underlying hearing characteristics and auditory adaptation in turtles.

**Abstract:**

An auditory ability is essential for communication in vertebrates, and considerable attention has been paid to auditory sensitivity in mammals, birds, and frogs. Turtles were thought to be deaf for a long time; however, recent studies have confirmed the presence of an auditory ability in *Trachemys scripta elegans* as well as sex-related differences in hearing sensitivity. Earlier studies mainly focused on the morphological and physiological functions of the hearing organ in turtles; thus, the gene expression patterns remain unclear. In this study, 36 transcriptomes from six tissues (inner ear, tympanic membrane, brain, eye, lung, and muscle) were sequenced to explore the gene expression patterns of the hearing system in *T. scripta elegans*. A weighted gene co-expression network analysis revealed that hub genes related to the inner ear and tympanic membrane are involved in development and signal transduction. Moreover, we identified six differently expressed genes (*GABRA1*, *GABRG2*, *GABBR2*, *GNAO1*, *SLC38A1*, and *SLC12A5*) related to the GABAergic synapse pathway as candidate genes to explain the differences in sexually dimorphic hearing sensitivity. Collectively, this study provides a critical foundation for genetic research on auditory functions in turtles.

## 1. Introduction

An auditory ability is necessary for most vertebrates to detect sound signals in the external environment and for their survival and reproduction, particularly for frogs, birds, and mammals [1,2,3]. Frequency sensitivity is a critical parameter of auditory characteristics that varies among species [4]. The hearing frequency range in humans is approximately 20–20,000 Hz [5]. However, other mammals have broader ranges than those of humans. For example, giraffes and elephants can communicate with infrasound (<20 Hz), whereas dolphins and bats can hear ultrasound (>20,000 Hz) [6,7,8,9]. Birds and anuran amphibians also have a wide range of hearing frequencies, including ultrasound [10,11]. In contrast, turtles and snakes respond only to low-frequency sounds. This led to the misunderstanding that these animals were deaf for a long time. However, the auditory function of turtles and snakes has been confirmed through combined behavioral and electrophysiological methods [12,13]. The electrical potentials in response to sounds have been recorded from the inner ear of many turtle species [14,15,16]. For *Chelonia mydas*, the ear is a low-frequency receptor with a probable range of 60–1000 Hz, which enables the animals to perceive several vital signals on land and in water [17]. Therefore, turtles are thought to have auditory functions that receive sounds within certain frequency limits.

The basic structures and function of auditory systems in vertebrates are identical [18]. The most well-developed auditory system structures are observed in mammals, for which the hearing apparatus consists of three parts: the outer, middle, and inner ear. The function of the outer ear is to collect and transfer sound energy to the middle ear [19]. The middle ear and functional hearing first appeared in amphibians [20], and some reptiles have a rudimentary external acoustic meatus [21]. The eardrum or tympanic membrane transforms sound into vibrations in the middle ear. The middle ear ossicles then amplify the signal before it is transmitted to the inner ear, where it stimulates the sensory receptors in the cochlea [22]. For turtles, the sound is mechanically conveyed to the inner ear by the tympanic membrane and middle ear ossicles [23]. The tympanic membrane and inner ear are two important sound receivers and processors in turtle auditory systems.

Auditory phenotypic diversity might be related to changes in gene expression. Significant efforts have been made to profile gene expression patterns in mouse inner ears at different developmental stages, as well as following various stimuli, physiological damage, and specific gene knock out [24,25,26,27]. Dong et al. (2013) reported that inner ear gene expression differs between echolocating bats and non-echolocating bats via a transcriptome analysis and identified several candidate genes related to echolocation ability [28]. Chen et al. (2022) compared differential gene expression in the brain transcriptomes of male and female concave-eared torrent frogs, *Odorrana tormota*, thus revealing the molecular mechanisms of differences in ultrasonic hearing between sexes [29]. Since the ear structure and hearing ability of turtles are different from those of other vertebrates, exploring the molecular architecture of the inner ear and tympanic membrane in turtles is warranted. Wang et al. (2019) measured hearing sensitivity in male and female *Trachemys scripta elegans* and observed that auditory brainstem response thresholds were significantly lower in females than in males for frequencies in the 0.2–1.1 kHz range, with the exception of 0.9 kHz, indicating that the hearing of females is more sensitive to this frequency [30]. However, the underlying molecular mechanism remains unclear.

Sexual dimorphism in hearing sensitivity is a great model to research the molecular basis of auditory functions in turtles. Therefore, we conducted comparative transcriptomics on six tissues (inner ear, tympanic membrane, brain, eye, lung, and muscle) to explore the molecular function related to hearing organs and to explain the difference in hearing sensitivity between sexes in *T. scripta elegans*. We believe that our study will provide novel insights into the genetic basis of auditory functions in turtles.

## 2. Materials and Methods

### 2.1. Collecting Samples

In this study, six 5-year-old *T. scripta elegans* individuals (three males and three females) were purchased from farms in Hainan Province, China (110.7191700° E, 19.8033100° N, 22 m). These turtles were anesthetized with Tricaine (MS-222) before euthanizing them. Six tissues (inner ear, tympanic membrane, brain, eyes, lung, and muscle) from each turtle were collected for RNA sequencing analysis. Specifically, the brain tissue refers to the whole brain, and the muscle tissue was sampled from the turtles’ forelimbs. All animal treatment procedures were approved by the Animal Research Ethics Committee of the Hainan Provincial Education Centre for Ecology and Environment, Hainan Normal University (HNECEE-2018-001).

### 2.2. RNA Extraction and Illumina Sequencing

Total RNA from each sample was extracted using TRIzol kit (Invitrogen, Carlsbad, CA, USA) following the manufacturer’s instructions. After RNA quantification, quality assessment, and purification, cDNA libraries were constructed following the method described by Zhu et al. [31]. After cluster generation, the library preparations were sequenced on an Illumina Novaseq 6000 platform, and 150bp paired-end reads were generated.

### 2.3. Transcriptomic Analyses

Clean reads were obtained by removing the adapter, poly-N, and low-quality reads. The read quality was verified using FastQC software. All clean reads were assembled by mapping them to the reference genome using the RefSeq assembly accession number GCF_013100865.1 [32]. Transcriptome data from this study were submitted to the Genome Sequence Archive (GSA, https://bigd.big.ac.cn/gsa/, accessed on 14 June 2022) under accession number CRA007204.

### 2.4. Weighted Gene Co-Expression Network Analysis (WGCNA)

WGCNA is a systematic biological method used to construct scale-free networks based on gene expression profiles. The potential tissue-specific gene modules were filtered using WGCNA v1.69 [33], which constructs a co-expression network of all genes based on transcripts per million (TPM). WGCNA can be used to find clusters of highly correlated genes, summarizing such clusters using the module eigengenes (MEs) or the most representative genes; to associate them with other modules and external sample traits; and to calculate module membership (MM) measures. MM was measured using Pearson’s correlation coefficient to analyze the correlation between MEs and traits. Gene significance (GS) and MM represent the correlation between gene expression profiles and MEs. The hub genes were screened to reveal the biological functions of the modules.

### 2.5. Screening of Differently Expressed Genes (DEGs)

DEGs were identified using DESeq2 (R package) [34] based on a threshold fold change > 2 (|log_2_Fold Change| > 1) and *q*-value < 0.05 (after Benjamini–Hochberg correction). We focused on the functional analysis of DEGs in the inner ear and tympanic membrane between males and females.

### 2.6. Enrichment Analysis of Gene Functions

A pathway enrichment analysis was performed based on the Kyoto Encyclopedia of Genes and Genomes (KEGG) and Gene Ontology (GO) databases using KOBAS 3.0 software [35]. The *p*-value was adjusted using the Benjamini–Hochberg false discovery rate, and terms with a corrected *p*-value < 0.05 were recognized as significant. GO enrichment analyses of genes in the inner ear and tympanic membrane-related modules were performed. A KEGG functional enrichment analysis was conducted for DEGs between males and females.

## 3. Results

### 3.1. Transcriptome Sequencing Analysis

In this study, we obtained 233.8 Gb of raw data from six tissues of six *T. scripta elegans* individuals (three males and three females; Appendix A and Figure 1). After quality control, a total of 221.4 Gb of clean reads was obtained for downstream analysis. Clean reads were mapped to the reference genome, and the mapping rates of the 36 samples ranged from 86.3% to 92.0% (Appendix A).

### 3.2. Co-Expression Network Construction

WGCNA was used to screen the characteristic gene groups in the tissues. According to the model, connectivity among genes in the network indicated a scale-free network distribution; as the correlation coefficient threshold was set to 0.9, the best soft-thresholding power was seven (Figure 2A,B), and modules with high ME correlations (R^2^ > 0.8) were merged (Figure 2C). We conducted a correlation analysis between the modules and tissues to identify the modules most relevant to auditory tissues. The results revealed that 16,398 genes could be divided into nine co-expression gene modules, and the blue (R^2^ = 0.53, *p* = 8 × 10^−4^, gene number = 2057) and red (R^2^ = 0.63, *p* = 4 × 10^−5^, gene number = 729) modules were significantly correlated with the inner ear and tympanic membrane, respectively (Figure 2D).

### 3.3. Functional Enrichment Analysis of Significant Modules

In the gene module of the inner ear, 200 hub genes (GS > 0.6 and MM > 0.6) were selected to reveal the biological functions of the inner ear (Figure 3A). The GO functional enrichment analysis assigned these genes to three categories: biological process, molecular function, and cellular component (Figure 3B and Appendix A). For the GO category biological process, the hub genes of the inner ear were mainly enriched in intracellular protein transport (GO:0006886), signal transduction (GO:0007165), and regulation of sodium ion transport (GO:0002028). These pathways are closely related to cellular signal transmission. Other highlighted biological processes included inner ear morphogenesis (GO:0042472) and neural crest cell development (GO:0014032), which are related to structural formation in the auditory system. For the GO category molecular function, the hub genes of the inner ear were enriched in potassium ion binding (GO:0030955), magnesium ion binding (GO:0000287), and calcium ion binding (GO:0005509), which are related to mechanoelectrical transduction. For the GO category cellular component, extracellular exosome (GO:0070062) and plasma membrane (GO:0005886) suggested cell activity in the inner ear. From significant pathways enriched in the inner ear, we selected four featured genes in these GO categories as hearing-related genes, as follows: *SOX9* (“signal transduction”， GO:0007165), *FGFR1* (“inner ear morphogenesis”， GO:0042472), *ATP1A1* (“potassium ion binding”， GO:0030955), and *IDS* (“calcium ion binding”， GO: 0005509). Notably, the expression of these four genes was higher in the inner ear than in other tissues (Figure 3C).

In the gene module of the tympanic membrane, 228 hub genes (GS > 0.5, MM > 0.5) were selected (Figure 4A). GO terms belonging to biological processes included signaling receptor binding (GO:0005102) and signal transduction (GO:0007165), indicating sound processing of the tympanic membrane. The others were associated with biological processes including inner ear development (GO:0048839) and skeletal system development (GO:0001501), which are related to development of the auditory system. Moreover, innate immune response (GO:0045087) and response to wounding (GO:0009611) suggested active immune responses in the tympanic membrane. The cellular components included ribosomes (GO:0005840), synapses (GO:0045202), and extracellular exosomes (GO:0070062), suggesting robust protein synthesis and secretion in this tissue (Figure 4B and Appendix A). From significant pathways enriched in the tympanic membrane, we selected four featured genes, as follows: *BMP5* (“negative/positive regulation of cell population proliferation”， GO:0007165/0,008,284 and “ossification”， GO:0001503), *RPL38* (“ossification”， GO: 0001503), *CCN2*, and *S100A11* (“protein binding”， GO:0005515 and “signal transduction”， GO:0007165). These four genes exhibited higher expression levels in the tympanic membrane than in other organs (Figure 4C).

### 3.4. Expression of DEGs and Screening of Genes in Pathways

Next, we compared the transcriptional differences in the inner ear and tympanic membrane between male and female *T. scripta elegans*. Twenty-three DEGs were identified in the tympanic membrane; however, this number is negligible and is not discussed (Appendix A). There were 153 DEGs in the inner ear identified, among which the expression levels of 135 and 18 genes were upregulated in males and females, respectively (Figure 5A and Appendix A). KEGG enrichment analysis revealed that these DEGs were mostly related to synaptic exaction and inhibition of neurons and neuronal signaling, including the GABAergic synapse, glutamatergic, and neurotrophin signaling pathways (Figure 5B). Notably, the GABAergic synapse pathway, which modulates nerve excitation, was significant among KEGG terms. In the GABAergic synapse pathway, six DEGs (*GABRA1*, *GABRG2*, *GABBR2*, *GNAO1*, *SLC38A1*, and *SLC12A5*) were identified, all of which exhibited higher expression levels in male *T. scripta elegans* than in females (Figure 5C). These genes play critical roles in synaptic activity, and their combined action leads to hyperpolarization and decreased excitability in postsynaptic cells (Figure 6).

## 4. Discussion

Currently, only limited studies exist on gene expression related to auditory functions in turtles. Here, we comprehensively probed the molecular functions of the inner ear and tympanic membrane of *T. scripta elegans* based on comparative transcriptomics and revealed that it is highly correlated with auditory organ development and signal transduction. We identified six candidate genes in the GABAergic synapse pathway, which were determined to be involved in differences in hearing sensitivity between sexes.

### 4.1. Molecular Functions in the Inner Ear

The inner ear is an indispensable component of the auditory system in vertebrates, and its primary function is to detect sound and to convey signals to the brain [36]. Our results indicated that *SOX9*, *FGFR1*, *ATP1A1*, and *IDS* were expressed at high levels in the inner ear. *SOX9* is an evolutionarily conserved transcription factor, and studies on zebrafish, African clawed frogs, chickens, and mice have demonstrated its essential function in the inner ear and cranial neural crest development [37,38,39,40]. Fibroblast growth factor receptor 1 (FGFR1), a tyrosine kinase receptor, is expressed in the inner ear of mice and is necessary for the development of the auditory sensory epithelium [41]. Functional enrichment analysis showed that *FGFR1* is involved in numerous biological functions, such as inner ear morphogenesis, protein binding, and calcium ion binding, which suggests that it plays a vital role in the function of the inner ear, specifically in receiving and processing sound, in *T. scripta elegans*. The *ATP1A1* gene encodes Na/K-ATPase α1 isoforms [42], which are critical for maintaining hyperpolarized membrane potential by controlling cellular sodium and potassium concentrations [43]. This indicates that *ATP1A1* is essential for mechanoelectrical transduction in the inner ear. Hair cells of the inner ear contribute to the initial steps in the neural processing of sound and balance in vertebrates [44]. *IDS* is an important gene in cochlea development in mice and chickens; further, it regulates hair cell differentiation and is expressed in the prosensory domains of the otic vesicle [45]. Overall, these four genes were confirmed to be associated with the development of auditory organs and sound transduction and are indispensable for the functional maturation of the inner ear in *T. scripta elegans*. These findings will allow us to further explore the molecular mechanisms underlying inner ear development and hearing in turtles.

### 4.2. Molecular Function of the Tympanic Membrane

The tympanic membrane is the fundamental “sound receiver”， as confirmed by audiograms of *T. scripta elegans* [23]. In addition to being a critical component of the auditory system that transmits sound to the inner ear, the tympanic membrane also participates in the defense system of the middle ear and can respond to immunological stimulation [46]. We identified significant pathways and genes implicated in the tympanic membrane. *BMP5* belongs to the transforming growth factor (TGF)-β superfamily [47] and is essential for the formation of particular skeletal elements and the development of several soft tissues [48]. Our results revealed a higher expression level of *BMP5* in the tympanic membrane than in the other five tissues, indicating that this gene is critical for the fundamental morphogenesis of auditory organs. Furthermore, *Rpl38* is indispensable during the development of the middle ear, and its deletion results in ectopic ossification and cholesterol crystal deposition in the middle ear cavity, an enlarged Eustachian tube, and chronic inflammation with effusion, all of which cause conductive hearing loss in mice [49]. *CCN2* encodes the connective tissue growth factor (CTGF) protein, a matricellular protein involved in many biological processes, including protein binding, cell proliferation and differentiation, and signal transduction [50]. The molecular and biological characteristics of *CCN2* might be crucial for the development and function of the tympanic membrane. S100A11, a member of the calcium-binding protein S100 family, might mediate signal transduction with various stimuli [51]. Moreover, *S100A11* also has biological functions, including the regulation of enzyme activity and inflammatory responses [52]. Our results collectively suggest that the tympanic membrane is vital for maintaining normal hearing functions in *T. scripta elegans*.

### 4.3. Molecular Mechanisms Underlying the Differences in Hearing Sensitivity between Sexes

Wang et al. [30] reported that female and male *T. scripta elegans* have a similar range of frequency sensitivities (0.2–0.9 kHz). However, the auditory brainstem response (ABR) threshold of males is significantly higher than that of females, suggesting that sexually dimorphic hearing sensitivity has emerged in turtles. In this study, numerous genes with different expression patterns between males and females in *T. scripta elegans* were identified, of which six auditory-related genes (i.e., *GABRA1*, *GABRG2*, *GABBR2*, *GNAO1*, *SLC38A1*, and *SLC12A5*) are involved in the GABAergic synapse pathway and might influence hearing sensitivity. In the mature central nervous system (CNS), γ-aminobutyric acid (GABA) is the chief inhibitory neurotransmitter, and it exerts its effects through ionotropic (GABAA and GABAC) and G protein-linked metabotropic (GABAB) receptors to achieve fast synaptic inhibition [53,54].

GABAA receptors are essential for the formation and development of GABAergic synapses [55]. *GABRG2* and *GABRA1* encode the GABAA receptor subunits α1 and γ2, respectively, and function in controlling neuronal excitability [56]. The functions of GABAA receptors depend on their expression levels, distribution patterns, and chloride (Cl^–^) transmembrane gradients. *SLC12A5* encodes potassium chloride cotransporter 2 (KCC2), which can mediate GABAergic ionic plasticity [57,58]. *GABBR2* encodes GABAB receptor subunit 2, and the GABAB receptor is a G protein-coupled receptor (GPCR) that is specifically associated with a subset of G proteins (pertussis toxin-sensitive Gi/o family) that modulate voltage-gated calcium channels (VGCCs). Activation of this receptor triggers the GTP-dependent release of G protein heterotrimers (Gα-GTP and Gβγ), which mediate the direct inhibition of vesicle release, along with indirect inhibition by the suppression of VGCCs [59,60,61]. *GNAO1* encodes a specific α-subunit of heterotrimeric guanine nucleotide-binding proteins (Gαo, a member of the Gi/o family), which are abundant in tissues of the mammalian CNS [62,63]. In addition, GIRK2, as a member of the G protein-gated inwardly rectifying potassium (GIRK) channel family, is also directly gated by G proteins, specifically the Gβγ dimer [64,65], and GIRK channel activation via Gi/o-coupled GPCR results in hyperpolarization of the neuron, thereby inhibiting neuronal activity [66,67,68]. Moreover, overexpression of the *KCNJ6* gene, which encodes the GIRK2 channel, can lead to enhanced GABAB receptor-mediated GIRK currents in mice [69]. Therefore, the upregulation of GABAB receptor subunit 2 (*GABBR2*) expression can regulate GABAB–GIRK currents, which are considered inhibitory. GABAc receptors (encoded by *GABRR3*) also play an important role in inhibiting postsynaptic cells [70]. Experiments in mice have demonstrated that solute carrier 38 member a1 (SLC38A1) plays a key role in GABA synthesis [71,72]. For male *T. scripta elegans*, these six upregulated genes were determined to be involved in the GABAergic synapse pathway (Figure 5C and Figure 6), which eventually leads to hyperpolarization and decreased excitability of the signals for neurons. When exposed to sound stimulation, GABAergic neurotransmission can balance the glutamatergic excitatory drive to control the neuronal output by providing the majority of synaptic inhibition. Therefore, the hearing of male *T. scripta elegans* might be insensitive compared to that of females, which is consistent with the phenomenon that males require a higher ABR threshold to hear. Our results reflect the differences in the differentiation and regulation of hearing-related neurons between males and females.

Revealing the differences in auditory sensitivity between males and females of *T. scripta elegans* provides not only better insight into the causes of greater hearing sensitivity in females but also an opportunity to explore the probable molecular mechanisms of sexual differences in auditory characteristics. Further analysis and verification are required to explore additional factors that might contribute to this difference in hearing sensitivity.

## 5. Conclusions

Our research complements the molecular details underlying auditory ability or at least the types of candidate genes and pathways involved. This study offers a new perspective to focus on the hearing characteristics and auditory adaptations in turtles.

## Figures and Tables

**Figure 1 animals-12-02410-f001:**
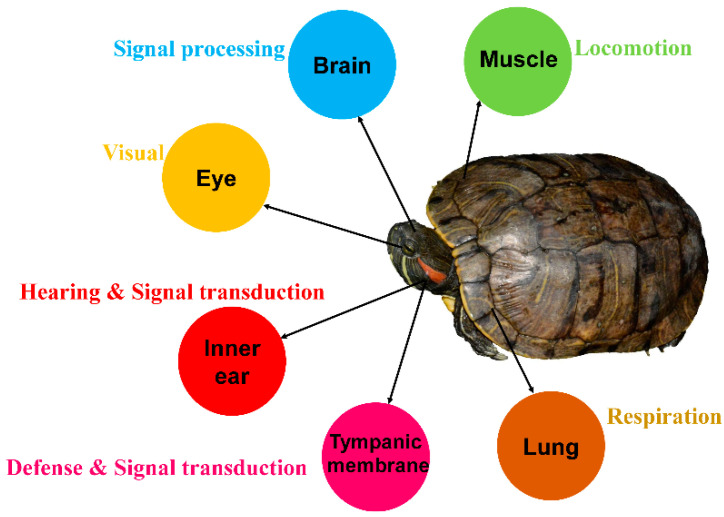
Biological functions of organs in *Trachemys scripta elegans*.

**Figure 2 animals-12-02410-f002:**
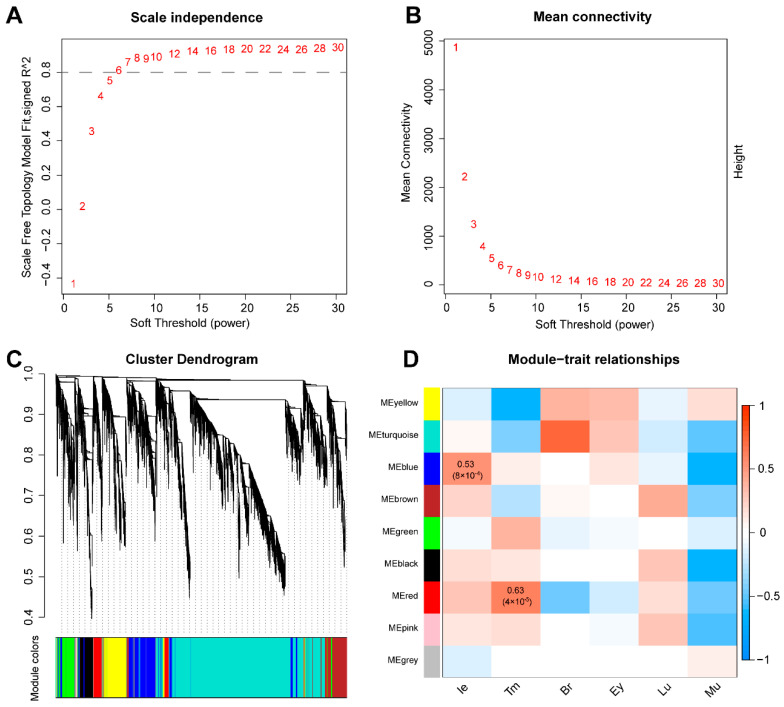
Co-expression analysis of all genes in the six tested tissues. (**A**,**B**) Analysis of soft-thresholding powers based on (**A**) scale independence and (**B**) mean connectivity. (**C**) Cluster dendrogram of genes. Modules with high similarity were merged. (**D**) Heatmap of correlation between modules and traits. The color scale shows the strength of correlation (Br: brain; Ey: eye; Ie: inner ear; Lu: lung; Mu: muscle; Tm: tympanic membrane).

**Figure 3 animals-12-02410-f003:**
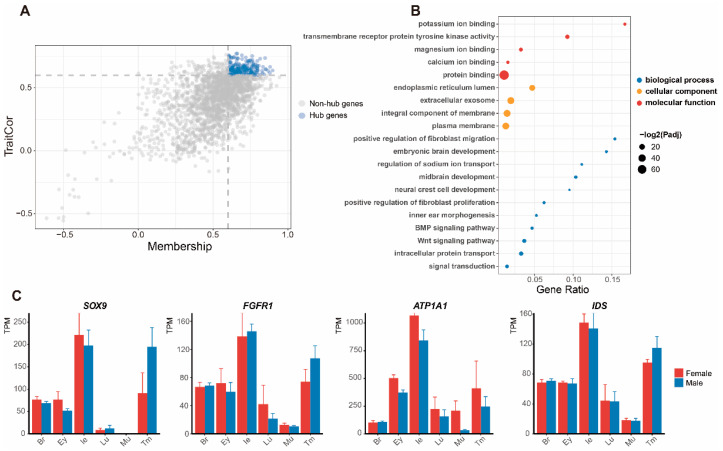
Screening and analysis of module genes associated with the inner ear. (**A**) Selection of hub genes in the dot plot. (**B**) Main Gene Ontology (GO) functional items enriched based on hub genes (Padj = corrected *p*-value). (**C**) Expression patterns of four featured genes in the six tissues.

**Figure 4 animals-12-02410-f004:**
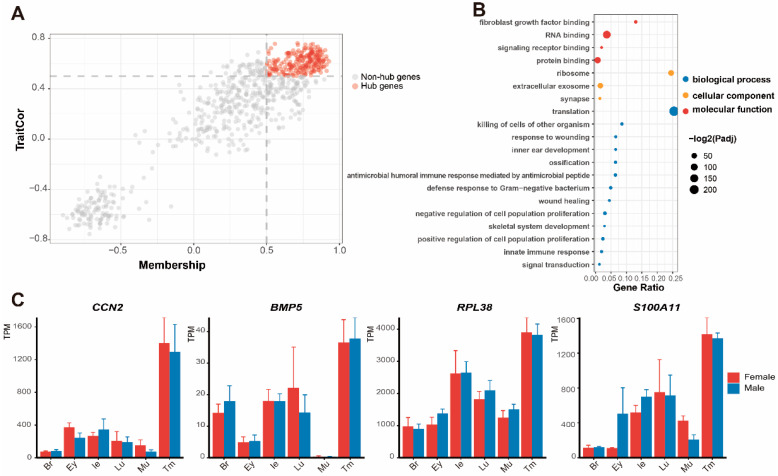
Enrichment analysis and expression of hub genes related to the tympanic membrane. (**A**) Selection of hub genes in the dot plot. (**B**) Main Gene Ontology (GO) functional items enriched based on hub genes (Padj = corrected *p*-value). (**C**) Expression patterns of four featured genes in the six tissues.

**Figure 5 animals-12-02410-f005:**
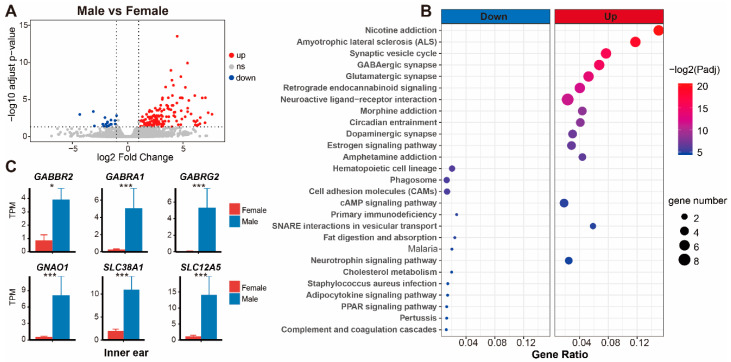
Comparison of transcriptional expression in the inner ear between males and females. (**A**) differentially expressed gene (DEG) screening in the inner ear between sexes. (**B**) KEGG functional items enriched based on DEGs (Padj = *q*-value). (**C**) Expression of six genes in the GABAergic synapse pathway. ***: *q* < 0.001, *: *q* < 0.05.

**Figure 6 animals-12-02410-f006:**
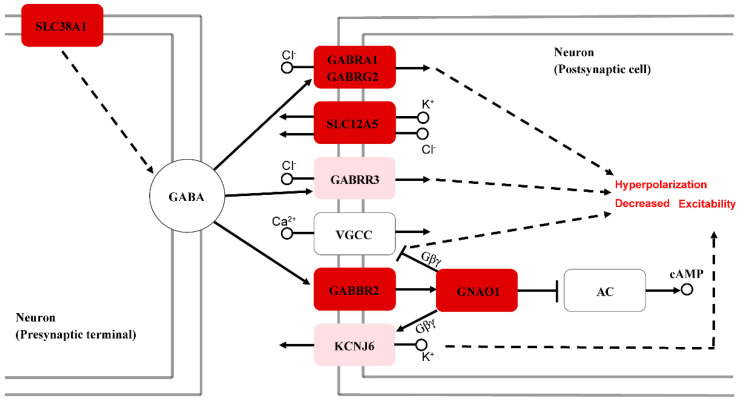
Regulatory mechanisms of six genes in the GABAergic synapse pathway (AC: adenylyl cyclase). The genes with high expression in males are shown in different shades of red boxes (fold change increases from light to dark red). The expression levels of genes between males and females in the light red boxes are shown in Appendix A.

## Data Availability

The sequencing data were submitted to the Genome Sequence Archive (GSA, https://bigd.big.ac.cn/gsa/, accessed on 14 June 2022) under accession number CRA007204.

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
