# Peer review of "Transcriptome Analyses Provide Insights into the Auditory Function in Trachemys scripta elegans"

_animals, 2022, doi:10.3390/ani12182410_

Round 1
Reviewer 1 Report
The authors do a nice job in studying the molecular basis underlying the auditory function of turtles and explaining the difference in hearing sensitivity between sexes. The transcriptomic data is well analyzed and successfully identified the critical genes and neural pathways associating with the hearing sensitivity. I have only several minor comments on the manuscript to make it to the best form.
1, The authors choose three functionally unrelated organs (lung, muscle and eye) as experimental samples, while these data are rarely mentioned in the results and discussion?
2, In the Weighted Gene Co-Expression Network Analysis, different thresholds were used to filter hub genes from modules related to the inner ear and tympanic membrane. What is the basis for choosing the thresholds? Please explain it in the Methods and Materials section.
3, In this work, the tympanic membrane DEGs were mentioned only for the numbers. Although these genes are few, I think they may still be valuable in function, so I suggest authors provide more information about them either in the results or in the discussion.
4, Please provide the statistical results (e.g., p values, q values, or asterisks denote these values) for the bar plots in the figures.
5, Line 86, 133 The font face of species name should be italic.
6, Line 87 Please provide the longitude and latitude of the farm.
7, Line 106 Please provide the GSA number rather than the Project number.
8, Figure 4b Please modify the text in Figures 3b and 4b to bold font face.
9, Figure 6 Please modify the text of Figure 6 as bold Arial.
Author Response
- The authors choose three functionally unrelated organs (lung, muscle and eye) as experimental samples, while these data are rarely mentioned in the results and discussion?
Reply: In this work, we focused on the molecular functions and differences between sexes based on the inner ear and tympanic membrane, because they are essential parts of the hearing system of T. scripta elegans. Meanwhile, in terms of the study design, control samples are also indispensable for analysis. Especially, for the weighted gene co-expression network analysis (WGCNA), certain irrelevant tissues are instrumental in screening genes that are specifically highly expressed in hearing organs. Thus, we finally chose these six organs for this study.
- In the Weighted Gene Co-Expression Network Analysis, different thresholds were used to filter hub genes from modules related to the inner ear and tympanic membrane. What is the basis for choosing the thresholds? Please explain it in the Methods and Materials section.
Reply: In WGCNA, it is necessary to obtain the module (gene set) associated with hearing organs. After that, screening genes from modules is optional, but this conformed to criteria regardless of whether genes were further filtered. Although this step is not indispensable, the parameters (module membership (MM) and gene significance (GS)) were effective to obtain hub genes that are more related to hearing organs. Both MM and GS are adjustable, and appropriate thresholds will be more helpful for the analysis. In this work, the module tympanic membrane had a higher correlation coefficient and a small-scale gene set relative to those of the inner ear. Thus, when we filtered the hub genes from modules, moderate thresholds (MM = 0.5, GS = 0.5) were chosen for the tympanic membrane, whereas stricter thresholds (MM = 0.6, GS = 0.6) were used for the inner ear.
- In this work, the tympanic membrane DEGs were mentioned only for the numbers. Although these genes are few, I think they may still be valuable in function, so I suggest authors provide more information about them either in the results or in the discussion.
Reply: Thank you for your helpful comments. The tympanic membrane is certainly an important organ for the hearing system. I have added information about these genes as a supplementary table (Table S6). However, based on the functional enrichment analysis of these genes, it is my opinion that they might have little effect on hearing functions based on different sexes, and thus, they have not been discussed in this study.
- Please provide the statistical results (e.g., p values, q values, or asterisks denote these values) for the bar plots in the figures.
Reply: Thank you for pointing this out. We have followed your suggestion and added these components to the manuscript.
- Line 86, 133 The font face of species name should be italic.
Reply: This has been addressed. Thanks.
- Line 87 Please provide the longitude and latitude of the farm.
Reply: This has been added. Thanks.
- Line 106 Please provide the GSA number rather than the Project number.
Reply: This has been changed. Thanks.
- Figure 4b Please modify the text in Figures 3b and 4b to bold font face.
Reply: This has been addressed. Thanks.
- Figure 6 Please modify the text of Figure 6 as bold Arial.
Reply: This has been performed. Thanks.
Reviewer 2 Report
The authors conducted an interesting study in identifying genes involved in hearing in turtles. My major concern is that it is not possible to statistically evaluate sex differences between 3 males and 3 females. That result is not defensible, nor are statistically significant values reported (lines 212-213). I recommend that the article is re-written to focus on the identification of genes and not on sex differences.
Add more detail to the Methods, so that every analysis is clearly described. Especially with statistical analyses, more detail could be added to the Methods.
Throughout the Results, define "gene modules" - this does not seem to be a general term in gene expression studies.
Some figures need additional edits/explanations. Figure 1 is not informative. Figure 2C: What are the module colors? 2D: What are the ME labels?
Figure 3D and 4D: there does NOT seem to be major differences between females and males based on this figure.
Author Response
- The authors conducted an interesting study in identifying genes involved in hearing in turtles. My major concern is that it is not possible to statistically evaluate sex differences between 3 males and 3 females. That result is not defensible, nor are statistically significant values reported (lines 212-213). I recommend that the article is re-written to focus on the identification of genes and not on sex differences.
Reply: Thank you for your helpful comments. Three biological replicates are credible for transcriptome analysis. When comparing sexes, three or fewer biological replicates are permitted when comparing auditory abilities via transcriptome analysis (Chen et al., 2022). The differential expression analyses between sexes were performed using DESeq2 based on the negative binomial distribution. Differentially expressed genes (DEGs) should meet the threshold of q < 0.05 after Benjamini and Hochberg’s correction. The significant DEGs in the inner ear (lines 212–213) have been listed in supplementary Table S5, but we followed your suggestion and addressed these issues in the manuscript. We have labeled the significant DEGs in the bar plot (Figure 5).
- Add more detail to the Methods, so that every analysis is clearly described. Especially with statistical analyses, more detail could be added to the Methods.
Reply: Thank you for your helpful comments. In accordance with your suggestion, we have modified the methods to provide these details.
- Throughout the Results, define "gene modules" - this does not seem to be a general term in gene expression studies.
Reply: The term “gene module” or “cluster module” is specific to WGCNA, and the more complete term is “co-expression module”, which is sometimes referred to simply as “gene module”, even in the original text describing the method (Langfelder and Horvath, 2008).
- Some figures need additional edits/explanations. Figure 1 is not informative. Figure 2C: What are the module colors? 2D: What are the ME labels?
Reply: Thank you for your helpful comments. Figure 1 shows the biological functions related to the six sampled tissues of the species. Figure 2C: The module color is used to differentiate various modules obtained through clustering, and the color is used as the label for each module throughout this study. 2D: “ME” is the abbreviated form of “module eigengene”. These have been specified in section 2.4, and we have made further improvements to this section to improve clarity.
- Figure 3D and 4D: there does NOT seem to be major differences between females and males based on this figure.
Reply: Figures 3D and 4D show the hub genes related to the inner ear and tympanic membrane, respectively. All genes shown in Figure 3D and 4D were highly expressed and have important molecular functions in the inner ear or tympanic membrane but are not necessarily differentially expressed between sexes. Meanwhile, Figure 5 illustrates the differential expression between sexes. Genes in Figure 5C were significantly differentially expressed in different sexes.
References
Chen, Z., Liu, Y., Liang, R. et al. Comparative transcriptome analysis provides insights into the molecular mechanisms of high-frequency hearing differences between the sexes of Odorrana tormota. BMC Genomics 23, 296 (2022). https://doi.org/10.1186/s12864-022-08536-2.
Langfelder, P., Horvath, S. WGCNA: an R package for weighted correlation network analysis. BMC Bioinformatics 9, 559 (2008). https://doi.org/10.1186/1471-2105-9-559.
Reviewer 3 Report
The authors did a transcriptomic analysis to study the difference in auditory sensitivity between male and female T. scripta elegans. The study is comprehensive but the authors should
1) show in a table the reproducibility of biological replicates for all six tissues to build confidence in the data.
2) The authors were able to identify DEG but failed to explain the role of these genes mechanistically during evolutionary development or the difference in hearing patterns of both sexes. A little more emphasis should be given to explaining the probable biology of these DEG.
Author Response
- The authors should show in a table the reproducibility of biological replicates for all six tissues to build confidence in the data.
Reply: Thank you for your helpful suggestion. All information about samples in this work is provided in supplementary Table S1, including biological replicates for all six tissues, sex-specific differences, and a statistical analysis of the RNA sequencing data. This information supports the contention that our study design and quality of sequencing data are credible.
- The authors were able to identify DEG but failed to explain the role of these genes mechanistically during evolutionary development or the difference in hearing patterns of both sexes. A little more emphasis should be given to explaining the probable biology of these DEG.
Reply: Thank you for your helpful comments. The biological functions of these differentially expressed genes are indeed important and worthy of emphasis. However, the genes known to be related to hearing ability in vertebrates were barely significant, in terms of DEGs between sexes. Moreover, there are various parameters used for measuring auditory ability, and the study was designed to reveal differences in auditory sensitivity between males and females of T. scripta elegans, which might have provided different evidence from that provided in previous studies. In our results, the GABAergic synapse pathway showed a difference between sexes. The expression levels of most DEGs in this pathway showed a similar trend to that of the entire pathway. Thus, an identified pathway with multiple genes provides confidence to explain the difference in hearing sensitivity, whereas the molecular function associated with a single DEG might not be capable of this. Thus, though no single gene has been emphasized, our overall results are also notable and might provide novel insights into the complexity of auditory functions.
Reviewer 4 Report
In this study the authors obtained the transcriptome of six tissues (inner ear, tympanic membrane, brain, eye, lung, and muscle) from Trachemys scripta elegans, and analyzed sex-specific differences related to auditory function. Six relevant DEGs were identified related to the GABAergic pathway with dimorphic expression in the inner ear, providing insights into sex-specific hearing sensitivity. The study is interesting, well written, and well presented, it provides relevant information regarding auditory function in turtles, using T. scripta as a model, nevertheless I still have some comments.
If the main goal was to study molecular function related to hearing organs, it is clear the inclusion of the inner ear, tympanic membrane, and brain, however the eye, muscle, and lung are not so obvious since they do not participate in the auditory function, at least not in a direct manner, so please explain the reason for including eye, muscle, and lung in this particular study.
In addition, be more specific regarding brain and muscle sampling procedure; indicate the exact sampling site within brain and muscle.
Although an Ethics Committee approved the study, briefly explain the protocols for animal transportation, maintenance, euthanasia, and tissue collection to ensure that animal welfare was respected.
In section 3.4, the authors mention “Twenty-three DEGs were identified in the tympanic membrane; however, this number is negligible and has not been discussed in this study…” Twenty-three DEGs may be a low number for screening the genes in pathways, but still could provide interesting information, sometimes small or subtle differences in gene expression are relevant. I suggest including the analysis as supplementary data with a brief discussion.
Finally, since this study compared male and female transcriptomes, probably the title should mention the sexually dimorphic nature of the study, and the conclusion should emphasize that hearing sensitivity might be sexually dimorphic.
Author Response
- If the main goal was to study molecular function related to hearing organs, it is clear the inclusion of the inner ear, tympanic membrane, and brain, however the eye, muscle, and lung are not so obvious since they do not participate in the auditory function, at least not in a direct manner, so please explain the reason for including eye, muscle, and lung in this particular study.
Reply: Thank you for your helpful comments. In this work, we focused on the molecular functions and differences between sexes in terms of the inner ear and tympanic membrane, because they are essential parts of the hearing system of T. scripta elegans. However, when designing the study, control samples are also indispensable for analysis. Especially, for the weighted gene co-expression network analysis (WGCNA), certain irrelevant tissues are instrumental in screening genes that are specifically highly expressed in hearing organs. Thus, we finally chose these six organs for this study.
- In addition, be more specific regarding brain and muscle sampling procedure; indicate the exact sampling site within brain and muscle.
Reply: Thank you for your helpful suggestion. We have modified the manuscript and provided more details for this as per your advice.
- Although an Ethics Committee approved the study, briefly explain the protocols for animal transportation, maintenance, euthanasia, and tissue collection to ensure that animal welfare was respected.
Reply: Thank you for your helpful comments. We have added these details according to your suggestion.
- In section 3.4, the authors mention “Twenty-three DEGs were identified in the tympanic membrane; however, this number is negligible and has not been discussed in this study…” Twenty-three DEGs may be a low number for screening the genes in pathways, but still could provide interesting information, sometimes small or subtle differences in gene expression are relevant. I suggest including the analysis as supplementary data with a brief discussion.
Reply: Thank you for your helpful suggestion. The tympanic membrane is certainly an important organ of the hearing system. I have added information about these genes as a supplementary table (Table S6). However, after functional enrichment analysis of these genes, it is my opinion that they might have little effect on hearing functions with respect to different sexes, and thus, they have not been discussed in this study.
- Since this study compared male and female transcriptomes, probably the title should mention the sexually dimorphic nature of the study, and the conclusion should emphasize that hearing sensitivity might be sexually dimorphic.
Reply: Thank you for your helpful comments. We focused on two components related to hearing in T. scripta elegans in this study. We conducted comparative transcriptomics to explore the molecular function related to hearing organs and explain the difference in hearing sensitivity between sexes. Therefore, we used the following title: “Transcriptome analyses provide insights into the auditory function in Trachemys scripta elegans” We think that this can cover both aspects and is concise.
Round 2
Reviewer 3 Report
The manuscript can be accepted in the present form.